# Machine Learning-Based Method for Detached Energy-Saving Residential Form Generation

Haixu Guo [1] , Ding Duan [1], Jincheng Yan [1], Keyuan Ding [1], Fengkui Xiang [2] and Ran Peng [1,*]

1    School of Civil Engineering and Architecture, Wuhan Institute of Technology, Wuhan 430074, China
2    China Gezhouba Group Co., Ltd., Wuhan 430022, China
\*    Correspondence: pengran@wit.edu.cn

**Abstract:** In recent years, machine learning has gradually been applied to building energy-saving designs to reduce the time consumption of the optimization screening stage. However, since most of the existing research scholars come from the fields of computers and engineering, the application of machine learning technology mostly involves complex programming as well as software in the field of engineering, which requires multiple software to be coupled to achieve. In view of the differences between disciplines and the high learning threshold, these theories are difficult to apply and promote in practical work in the field of architecture. In this regard, this paper focuses on the improvement of methods, based on the Grasshopper platform, proposes a detached energy-saving residential form generation design method and process, to explore the optimal energy-saving building form in a more concise and efficient way. Based on this new method, on the basis of verifying its feasibility through a residential building case, two machine learning algorithms, neural network (ANN) and support vector machine (SVM), are compared and studied, and the applicability of these two algorithms in different building performance indicators is further discussed. The results show that the ANN model has the highest accuracy and is more suitable for the prediction of building energy consumption; in view of the simple and fast operation of SVM, it is more suitable for comfort prediction with relatively low accuracy requirements. By combining the above two machine learning methods, work efficiency can be improved while satisfying the prediction of relevant performance indicators. This method can help architects quickly search for the best building energy-saving form design scheme in the scheme design stage and provide data support and information feedback for architects in design conception and deepening.

**Keywords:** detached house; energy efficient building; machine learning; multi-objective optimization; Grasshopper

## 1. Introduction

With the rapid growth of China's economy after the reform and opening up, residential buildings have flourished in the past decades and become one of the most important building types for national construction development and solving people's livelihood problems [1]. Residential buildings consume a large amount of primary energy, increase a large amount of carbon emissions, and become an important cause of global warming. Therefore, low-carbon residential construction is the most important task to achieve the goal of "double carbon".

In recent years, the multi-objective optimization method has been more widely used in the retrofitting and generation of energy-efficient buildings. Di Wu et al. [2] incorporated building cost into the optimization index and used the Design Builder software platform to optimize a comprehensive screening of building energy consumption and economics relying on the NSGA-II algorithm. Shao Teng et al. [3] took the regional climate as the entry point and used the particle swarm algorithm (PSO) and Hooke-Jeeves algorithm to work together with the coupling of multiple software such as GENOPT and Energy-Plus to carry out the design optimization study of the building envelope as a variable.

Zhang Yong et al. [4] incorporated comfort into the optimization index and proposed a decomposition-based multi-objective evolutionary optimization (MOEA/D) method for building energy efficiency design, which is comparatively better than the typical NSGA-II algorithm in terms of distributivity and convergence, and raised the issue of how to combine machine learning to reduce computational time consumption in the outlook. Yuan Gao et al. [5,6] relied on the Grasshopper platform to find the optimal design for winter heating energy consumption and comfort with building planning and single-unit parameters as variables, and incorporated the TOPSIS comprehensive evaluation method to filter the Pareto solution set. Chingwen Xue et al. [7] based on MATLAB platform with the help of C programming language, through the building performance of 1000 samples simulations and used neural networks (ANN) to establish mapping relationships between variables and performance parameters to speed up the convergence of the optimization screening phase.

Through combing the literature, in the study of multi-objective optimization of buildings, in order to achieve better convergence and distribution, most research focuses on the improvement of algorithms. However, since the optimization process requires the intervention of performance simulation software to simulate the corresponding building variables and then the performance parameters obtained are optimized and filtered, taking the Honeybee plug-in as an example, its simulation time is 1–20 min per time. In the process of multi-objective optimization, hundreds or thousands of performance simulations are often required, so most of the time in the optimization process is consumed in the performance simulation, which can take up to several hours. It can be seen that it is difficult to solve the above problem by algorithmic improvement alone.

Although the newly proposed agent model idea [8–12] reduces the time consumption in the performance simulation link through machine learning, it is mostly used for building energy consumption prediction and building reconstruction, and there is little research on the building form generation link. In addition, most of the scholars in the current research come from the computer and engineering fields, so the application of machine learning technology mostly involves complex programming and software in the engineering field and needs to be realized by coupling multiple software. In view of the differences between disciplines and the high learning threshold, these theories are difficult to apply and popularize in practical work in the field of architecture.

This paper focuses on the improvement of the method and takes Rhino-Grasshopper (GH), a popular parametric design software in the field of architectural design, as the operation platform, and adds innovative modules of data sampling, performance simulation drive, and accuracy evaluation in the Octopus plug-in through visual programming and improves the optimization algorithm and process equipped with it from the perspective of architects. The optimization algorithms and processes are improved to achieve a set of independent energy-efficient residential form generation design methods that are simple to operate and can be completed on only one software platform, which successfully solves the above problems. The applicability of two machine learning methods, ANN and SVM, for different prediction target values is further analyzed. The method can help architects quickly find the optimal energy-efficient building form design in the schematic design stage and provide data support and feedback to architects in design conceptualization and refinement.

## 2. Literature Review

As mentioned in the introduction of this paper, the application of machine learning in building energy consumption was originally used to solve the problem of excessive time consumption in the multi-objective optimization stage. Therefore, this section, in order to improve the efficiency of the multi-objective optimization problem for the origin, from algorithm improvement to the intervention of machine learning algorithms, then reviews the literature on the application of machine learning in building energy consumption and building generation, so as to analyze the advantages and disadvantages of existing

technologies and methods, so as to help explain the process of the research problems proposed in this paper and the value and significance of the research methods proposed in this paper.

### 2.1. Optimization

As an important branch of artificial intelligence, the theoretical prototype of optimization algorithms, "heuristic search", can be traced back to Herbert Simon's "The Sciences of the Artificial" book in 1969 [13]. With the increasing demand for design optimization in architectural design, more and more optimization algorithms are embedded in the architectural design platform, which provides a variety of available algorithms for architects to solve different design optimization problems. In recent years, multi-objective optimization algorithms have also been widely used in rural reconstruction. The research can be divided into the following two directions: the first is the improvement of the optimization algorithm, and the second is the optimization results and information feedback (Table 1).

**Table 1.** The development of optimization algorithm research problems.

| Research Direction | Research Problem | Reference |
|---|---|---|
| Improvements to the Optimization Algorithm | Galapagos plug-in based on genetic algorithm and annealing algorithm, but only for single-objective optimization. | [14] |
| | Octopus plug-in based on GH platform, this plug-in introduces SPEA-2 and Hyper-Volume Evolutionary Algorithm (HypE), which can perform multi-objective optimization compared to the previous generation Galapagos plug-in. | [15] |
| | The application of NSGA-II algorithm in Design Builder, MATLAB, and other software platforms is better than SPEA-2 algorithm in terms of convergence and distribution. | [2] |
| | The MOEA/D algorithm uses the information of adjacent problems to update the individual position, avoiding the population falling into local optimum, and has great advantages over the previous generation algorithm in maintaining the solution distribution. | [16,17] |
| | The introduction of machine learning solves the time-consuming problem of multi-objective optimization by establishing a "surrogate model". | [8–12] |
| Optimization results and information feedback | The optimization results appear in the visual form of the Pareto solution set, which is convenient for the architect to conduct intuitive screening, but the number of the generated solution sets is often large, which brings certain difficulties for the architect to analyze the optimization results. | [18,19] |
| | By grouping and clustering the distribution of design variants in the result space, architects can compare only the differences between different clusters, and avoid redundant information from analyzing a large number of similar design variants. | [20] |
| | The K-means clustering algorithm is used to group the design variants in the optimization results in the target result space, and the performance characteristics of each group are analyzed and compared through parallel coordinates. | [21] |
| | The Pareto solution set is further evaluated and screened by the TOPSIS comprehensive evaluation method, which reduces the workload of architects in the solution set screening process. | [5] |

The improvement of the optimization algorithm helps to improve its convergence and distribution. The algorithm that first appeared on the GH platform was the Galapagos plug-in developed by David Rutten [14]. The plug-in includes the genetic algorithm and the annealing algorithm. Among them, the genetic algorithm is based on the standard genetic algorithm and controls the genetic similarity of the two parent individuals selected in the hybridization process through the "breeding" parameter; the appearance of this plug-in

is also an important step in the era of computer optimization for architectural design. It is widely used in building performance optimization and urban climate improvement. However, since Galapagos can only be optimized for a single objective, Vierlinger et al. [15] developed an Octopus plugin capable of multi-objective optimization. The plug-in introduces SPEA-2 and Hyper-Volume Evolutionary Algorithm (HypE) to allow users to optimize multiple objectives and visualize the individuals found during the optimization process through a 3D coordinate system, which can help architects filter suitable design variants in the Pareto solution set. The emergence of Octopus has a greater improvement than Galapagos in terms of algorithm improvement and software interaction design. With the introduction of the NSGA-II algorithm in software platforms such as Design Builder and MATLAB, its comprehensiveness is better than SPEA-2. To improve, Wu Di [2] and other scholars use this algorithm to screen passive energy-saving technologies and comprehensive energy-saving technologies by means of multiple software couplings, but there are still problems such as slow convergence speed and easy local convergence. Then the MOEA/D (multi-objective evolutionary algorithm based on decomposition) algorithm appeared, which has great advantages in maintaining the solution distribution [16,17]; at the same time, by using the information of adjacent problems to update the individual position, it avoids the population falling into a local optimum. Zhang Yong [4] and other scholars established a building model in SketchUp, ran the MOEA/D algorithm in MATLAB to generate a new solution, and then used the Visual C++ interface program to decode the new solution to Energy-Plus and output the energy of the building. The index value of time consumption and uncomfortable time is proposed, and in its outlook, it proposes how to combine machine learning and other methods to solve the problem of computational time consumption.

Although the improvement of the new algorithm can improve the optimization efficiency to a certain extent because the performance simulation software is required to intervene in the optimization process to simulate the corresponding building variables, and then the obtained performance parameters are optimized and screened, so most of the time it will be simulated by the performance. It is difficult to solve the above problems by the improvement of the algorithm alone; with the development of computer technology in recent years, machine learning algorithms have been introduced into the multi-objective optimization design of buildings; that is, by establishing a "surrogate model" to reduce the performance in the optimization process. The time consumed by the simulation will be discussed in detail in Section 2.2.

At the level of optimization results and information feedback, the processing and visualization of optimization results are also important means to help architects extract design information from them. In the results of multi-objective optimization, they often appear in the form of Pareto solution sets. However, the number of generated solution sets is It is often larger, which brings certain cognitive difficulties for architects in analyzing and optimizing results [18,19], so the processing of solution sets is particularly important. Some researchers propose to post-process the optimization results; that is, to extract a small number of representative design variants from the optimization results containing a large number of design variants to help architects quickly grasp the core information reflected in the optimization results. For example, scholars such as Negendahl [20] carried out a design optimization experiment of building a facade curtain wall with energy consumption, sunshine, cost, and heat load as multiple objectives. In order to extract key information from the Pareto front surface of design variants, the authors chose to visualize the optimization results in an "objective space" composed of a coordinate system and group and cluster these design variants according to their distribution in the resulting space. Through clustering, architects can compare only the differences between different clusters and avoid redundant information from analyzing a large number of similar design variants. Then, Chen [21] and other scholars took the space cooling system of the building as the design optimization object and carried out multi-objective optimization for electricity consumption, cost, and

sunshine. The target result space is grouped, and the performance characteristics of each group are analyzed and compared by parallel coordinates.

In addition, scholars such as Gao Yuan [5] further evaluated and screened the Pareto solution based on the GH platform through the TOPSIS (technology for order preference by similarity to an ideal solution) comprehensive evaluation method, which reduced the workload of architects in the process of solution selection.

### 2.2. Application of Machine Learning in Architectural

In recent years, with the development of computer technology, digital design has gradually become the mainstream of architectural design, and the mutual penetration between disciplines has also brought the application of many new technologies to architectural design. Machine learning technology originated from artificial intelligence and statistics, and is widely used in image recognition, manufacturing, and other fields. In recent years, it has been used in the field of architectural design. Literature review according to 5 main directions of application of machine learning in architectural design (Table 2).

**Table 2.** The development of research problems in the application of machine learning in architectural design.

| Research Direction | Research Problem | References |
|---|---|---|
| Building Performance Prediction | The time-consuming problem in multi-objective optimization is solved by establishing a "surrogate model". | [8–12,22] |
| | Based on the Rhino platform, a real-time visualization modeling platform is established by using GAN neural network, which can help architects to analyze the impact of urban wind and heat environment intuitively. | [23] |
| | Generating 3D Wind Field Data Based on SOM Algorithm. | [24] |
| | Based on the idea of "surrogate model", scholars such as Thomas Wortmann developed an optimization plug-in (Opossum) based on the GH platform. Compared with the traditional optimization algorithm, this plug-in can obtain performance optimization results more quickly. | [25] |
| Architectural Form Generation | Based on the GH platform, a three-dimensional convolutional neural network (3dCNN) is created. By transferring the three-dimensional shape of the building, the network can identify three architectural features. | [26] |
| | On the basis of the former research, the adjacency information of each sampling point is added, and the translated building shape information is handed over to the autoencoder for processing. | [27] |
| | Compared with the previous generation method, the three-dimensional model of the building is segmented in different directions, and the image of the cut surface is used as the learning information of the input of the neural network, and then the GAN neural network is used to generate the building form. | [28] |
| Building Plan Generation | Generative Design Method Based on GAN Neural Network. | [29–31] |
| | The machine learning method based on graph structure (Top-view representations) is a kind of deep learning algorithm. Compared with the GAN neural network algorithm, this method has more advantages in expressing the spatial topology relationship of buildings. | [32–35] |
| Building Renovation | Building facade generation based on GAN neural network. | [36] |
| | Building Performance Improvement Based on ANN Neural Network. | [7] |

Machine learning has demonstrated its potential as a "pattern recognition" tool for applied research in building performance prediction. Among them, in solving the problem of slow convergence in the multi-objective optimization stage mentioned in Section 2.1, the application of machine learning technology in this stage is mainly used to establish the relationship between input and output variables, also known as the surrogate model; that is, the machine learning algorithm learns the relationship between input and output

variables from multi-dimensional sample data, so as to establish a model of the correlation between input and output values, so this method is widely used in building energy consumption prediction [8–12], architectural environment simulation [22] and other building performance related fields. In solving the urban-scale environmental performance simulation, Duering S, Chronis, and other scholars established a real-time visualization modeling platform based on the Rhino platform using the GAN neural network [23]. The relationship between the graphs can help architects to intuitively analyze the influence of urban wind and heat environment, thereby optimizing the urban form. In addition, the SOM algorithm also has corresponding applications in generating 3D wind field data [24]. Based on the idea of a "surrogate model" in machine learning, scholars such as Thomas Wortmann [25] developed an optimization plug-in (Opossum) based on the GH platform. Compared with the traditional optimization algorithm, the plug-in can obtain performance optimization results more quickly, but due to the random number sampling method being used, results in a low model accuracy.

At the level of architectural form generation, in recent years, the focus has been on how to transfer the 3D shape of the building for machine learning to call, and it has been applied in the 3D convolutional neural network (3dCNN), that is, the 3D model of the building is pixelated. By making a three-dimensional distribution of pixels, the corresponding neural network can be created. At present, David Newton [26] created a three-dimensional convolutional neural network based on the GH platform, which can identify three architectural features, and then scholars such as Jaime de Miguel [27] added the adjacency information of each sampling point on this basis, The translated architectural shape information is handed over to the autoencoder for processing, and the architectural shape information is finally compressed into a vector in a latent space through four hidden layers. Scholars such as STEINFELD [28] adopted another method to establish a method of building shape generation based on machine learning. That is, another idea was adopted in the translation of building shapes, and the three-dimensional model of the building was divided in different directions. The image is used as the learning information at the input of the neural network, and then the GAN neural network is used to generate new architectural shapes.

The research on building plan generation is mainly divided into two different paths. The first is the generative design method based on the GAN neural network, that is, training in the form of image pixels to obtain a new building plan scheme [29–31]. In recent years, this method has also been widely used in the reconstruction and generation of building facades, but the problem is that it cannot be accurately exchange data with modeling software. The second is a machine learning method based on top-view representations, which is a kind of deep learning algorithm. This method has more advantages than the GAN neural network algorithm in expressing the spatial topology relationships of buildings [32–35].

With the saturation of urban buildings nowadays, building renovation gradually becomes a new research hotspot, and machine learning techniques are applied here. Thanks to the advantages of GAN neural networks in processing two-dimensional image data, similar to building planes, GAN neural networks are mainly used in the study of building façade renovation, which can assist architects in rapid batch pattern design in old neighborhoods and township renovation [36], eliminating the need to manually perform a large number of design patterns based on "mechanical" operations. In building performance retrofitting [7], neural networks (ANNs) are widely used because of their outstanding performance in terms of model accuracy, mostly using the MATLAB platform to invoke performance simulation software through programming, and then a certain number of simulation data samples are handed over to neural networks for learning to establish "agent models". Finally, the best combination of building parameters is selected through a multi-objective optimization algorithm, which can also be regarded as a further application of machine learning technology in the area of building energy consumption.

Through the review of related research status and problems and the reviews of "optimization algorithm" and "application of machine learning technology in architectural design", it can be seen that machine learning technology has been widely used in architectural design. Although SVM, ANN, and other traditional machine learning algorithms do not have the same "magic" as deep learning, they have great potential in the fields of building performance and building renovation, and the performance feedback in multi-objective optimization is accelerated by machine learning technology, which is also more in line with "intelligent" building needs. However, since most of the scholars in the existing research are from computer and engineering fields, the application of machine learning techniques mostly involves complex programming and software in the engineering field and requires multiple software coupling to achieve, which is difficult to promote in the practical work in the field of architecture due to the difference between disciplinary fields and high learning threshold.

## 3. Detached Energy-Efficient Housing Generation Method

In contrast to the diversity and complexity of urban architecture, detached housing is often characterized by modality and lightness, and passive energy efficiency tools such as form adaptation are particularly important. With the development of computer technology, the relationship between computers and architecture has become closer and closer. From the early days when the two fields were unrelated, to the 1990s when architects began to use computers for simple drafting and modeling, and nowadays when computers begin to assist architects in simple design tasks, architectural design has gradually differentiated into an independent research system.

Building design generation as a broad computer-generated concept covers many building generation ideas and methods. Specifically, it can be divided into the following two categories: the first is the "top-down" generation approach, i.e., the design generator builds a parametric model and then hands it over to the solver for filtering using optimization algorithms. This method is usually used in cases where there are few design elements and the relationship between them is simple, such as residential buildings with few building blocks and simple functions, or the generation of building façade components; the second type is the "bottom-up" operational design optimization, which integrates the solution and generation process of the design problem into a unified algorithm, which can solve the layout problem among multiple buildings, but the optimization algorithm is difficult to intervene [13–15]. Therefore, it is appropriate to adopt the first type of "top-down" approach in this study.

This paper regards building energy conservation as an important index of building form generation. The genetic algorithm carried in Octopus is used to optimize and screen the energy conservation indicators of buildings, so as to obtain the building form with the lowest energy consumption and relatively high comfort. As mentioned in the introduction, in the previous solution process, this stage often took a lot of time, and the intervention of the machine learning algorithm greatly shortened the time consumed in this process.

The parametric model part mainly includes parameter setting, LHS, performance simulation, machine learning, and accuracy evaluation, which is the core part of this study; the optimization algorithm part uses the Octopus solver for optimization screening. The detailed process is shown in Figure 1.

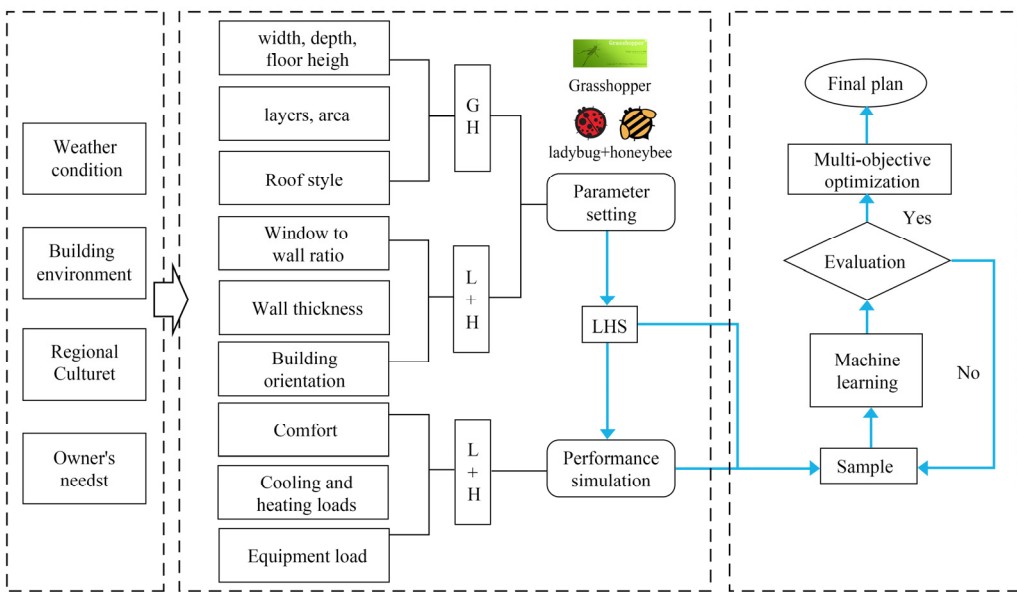

**Figure 1.** Flow chart of generation and optimization of detached energy-saving houses.

### 3.1. Parameter Setting

In the parameter setting, the architect needs to use field research and environmental monitoring to analyze the meteorological conditions, cultural customs, the owner's needs, and building codes of the countryside to set the number, type, and value range of the corresponding variables. The basic form parameters such as width, depth, height, and area of the building can be controlled by native commands such as domain box and number slider in GH; the parameters such as window to wall ratio and building orientation need to be controlled by the corresponding battery in the create module of Honeybee.

### 3.2. Latin Hypercube Sampling

Honeybee is a full-featured and highly accurate performance simulation plug-in based on the GH platform. Its internal operator can directly invoke the Energy-Plus calculation kernel to complete simulation analysis of annual thermal and cooling loads, comfort, etc. It is important to note that there are usually hundreds of thousands of variable parameter values arranged and combined in buildings, and as mentioned in the introduction of this paper, it would take a long time to perform computational simulations if Octopus is used directly for optimization screening at this stage. Therefore, it is necessary to sample the parameter values to generate a small and representative sample, and then input the sample variables into Honeybee separately for performance simulation, and finally establish the relationship between the design variables and the target values by means of machine learning.

Latin hypercube sampling (LHS) is a special stratified sampling method [37,38]. Its characteristic is that the collected sample data has good distribution and avoids the problem of sample data aggregation caused by the traditional random sampling method. Therefore, it is also widely used in computer experiments. However, due to the lack of plug-ins for sample collection in octopus, and the existing studies mostly using Python or R language programming to sample [6,8], the operation is cumbersome and requires other software intervention. Therefore, this paper supplements this cluster in GH through visual programming (Figure 2).

With a total of m variables $x_1$, $x_2$, ... $x_m$, N samples need to be collected. According to the sampling principle of LHS can be summarized as the following three steps: segmentation, taking values, and disruption. Step 1 (Segmentation): Use Divide Domain in GH to decompose each of the m variables into the same N small intervals. The number of small intervals is the number of required samples. Step 2 (take value): Use the Random

command to select a random value in each interval. If you want to adjust the number of decimal places to be retained in this stage, you can enter the corresponding expression in Evaluate. Step 3 (Disorder): Finally, use the Jitter command to disorder each of the N values in the m variables and then combine them with the values in the other variables.

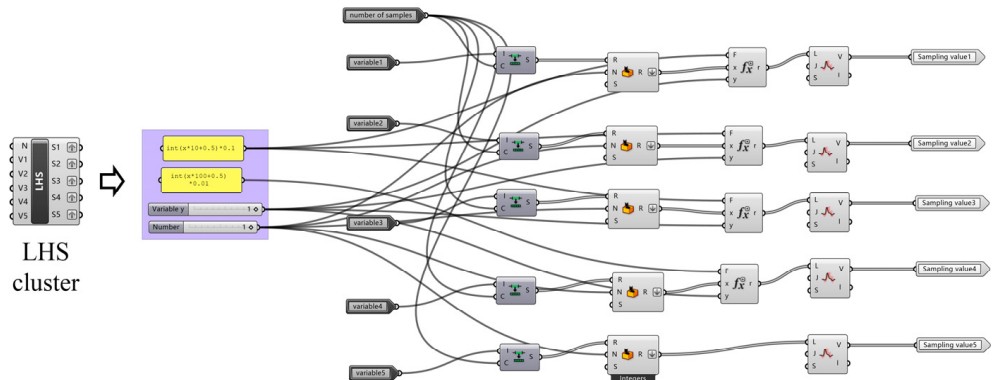

**Figure 2.** The battery diagram of the Latin hypercube sampling method based on GH.

### 3.3. Performance Simulation

Before the performance simulation, we need to clarify the performance target; generally, the more target values need to be simulated, the longer it takes. Relying on the power of Honeybee, the target values can be obtained through simulation calculation, such as cooling and heating load, equipment load, comfort, solar radiation value, and other performance targets, which can be set flexibly according to the demand in specific project practice. After setting the performance targets of the building, it is necessary to import the parameter samples obtained by LHS sampling into Honeybee for performance simulation, but the number of samples is often hundreds and it is difficult to perform manually, so it is necessary to build a driver module in GH that can import the data from the sample library into Honeybee simulation in turn. In this paper, the simulated parameters in the sample data are exported sequentially by data record and timer commands, and the parameter values to be simulated are assigned to the corresponding cell groups sequentially by list item (Figure 3). When all the parameter samples are simulated, the simulated values collected in the data record are checked and verified for later use in machine learning.

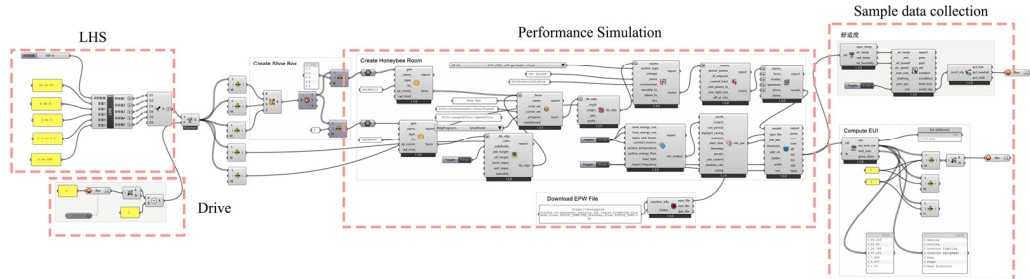

**Figure 3.** Performance simulation computing battery pack.

### 3.4. Machine Learning and Evaluation

#### 3.4.1. Machine Learning

Machine learning originated in the fields of artificial intelligence and statistics. In recent years, it has been used in the field of building energy consumption prediction to speed up optimization. That is, an input-output correlation model is established through machine learning, and new data is input into the trained model. Predictive models can predict the corresponding data outputs, a process known as "surrogate models" or "metamodels" [39].

As mentioned in Section 2.2, there are many types of machine learning algorithms; however, due to the different characteristics and principles of the algorithms, they have

their own areas of expertise; for example, the GAN neural network in deep learning is good at identifying image data, so it is mostly used in the field of architecture for the generation of plans and facades. In this study, the predicted indexes are the performance parameters of detached houses, including building energy consumption and comfort, which are regression problems with data labels, and the application of traditional machine learning algorithms has more potential in the field of building energy consumption and renovation, therefore, this paper unfolds with ANN and SVM learning algorithms.

In addition, relevant machine learning plug-ins have appeared in GH in recent years, such as Lunchbox, Octopus, Owl, and other plug-ins that package algorithms into different operators, among which Octopus is equipped with machine learning and multi-objective optimization modules, which have the characteristics of simple operation and relatively comprehensive functions; therefore, it is appropriate to choose Octopus as the operating platform for the machine learning part.

An ANN is a mathematical model that simulates a biological neural network for information processing, referred to as a neural network. It has the characteristics of high classification accuracy, strong learning ability, and good prediction and classification ability for untrained data [40].

The emergence of the MP-neuron model laid the foundation for the development of neural networks, however, the model with only one layer of functional neurons has limited learning ability and can only solve linearly separable problems for the nonlinear problem in Figure 4a cannot be fitted accurately. The neural network composed of multiple neurons and hidden layers enables ANN to deal with nonlinear problems better (Figure 4b). As shown in Figure 4, the greater the number of neurons, the smoother the fitted curve and the higher the accuracy of the model, but if the number of neurons exceeds a certain value, overfitting will occur and the accuracy decreases. In the problem of building performance prediction, the output values of energy consumption and indoor comfort are continuous variables, which can be regarded as a regression problem. The neural network can build a prediction model with high accuracy by adjusting the number of hidden layers and the number of neurons, so it is widely used in the field of building performance prediction and is therefore one of the machine learning algorithms focused on in this paper.

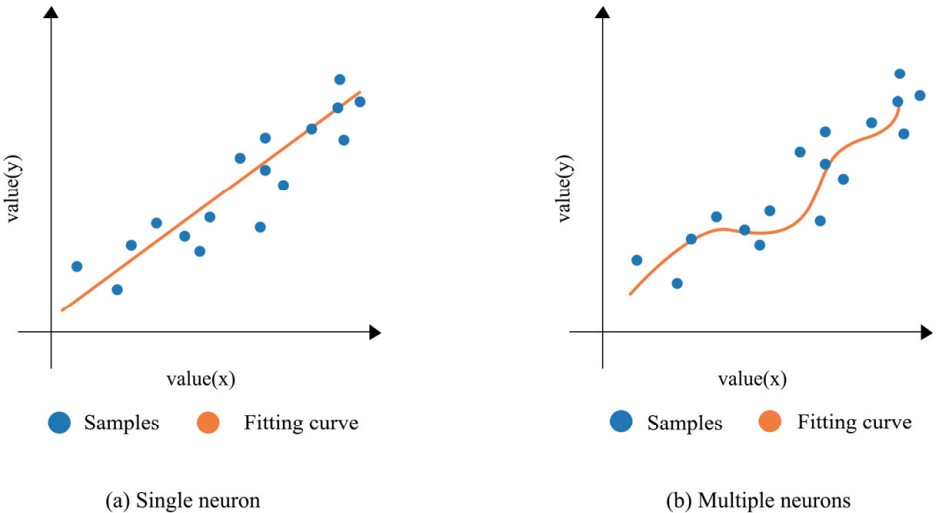

(a) Single neuron　　　　　　　　　　　　　　　　(b) Multiple neurons

**Figure 4.** Neural network fitting curve comparison.

Its operation in Octopus relies mainly on the following two parts: the input module (network learning) and the output module (network evaluate). The input of the learning module corresponds to the sample parameter values obtained from LHS; the output corresponds to the target values after simulating the performance of the sample parameter values; since the input and output values in this battery module only identify the values within $[-1, 1]$, the sample data needs to be mapped to the interval $[-1, 1]$ and then given

to network learning for neural network training. The infrastructure of a neural network is composed of multiple layers of parallel units called neurons, so the number of hidden layers and the number of neurons inside are important factors that affect the accuracy of a neural network, and if the neurons are underestimated, the ANNs ability to store information may be reduced. On the contrary, overestimation may lead to unnecessary learning and even overfitting of the neural network, and the "trial-and-error" method is commonly used to find the right number of hidden layers and neurons [7]. In Octopus, the number of hidden layers and neurons can be adjusted by controlling the parameter values of layers and nodes at the input side of the network learning battery. After the training is completed, the trained neural network is connected to the network evaluate battery, and the target value after prediction through the neural network is obtained by inputting the parameter values that need to be simulated on the I side (Figure 5).

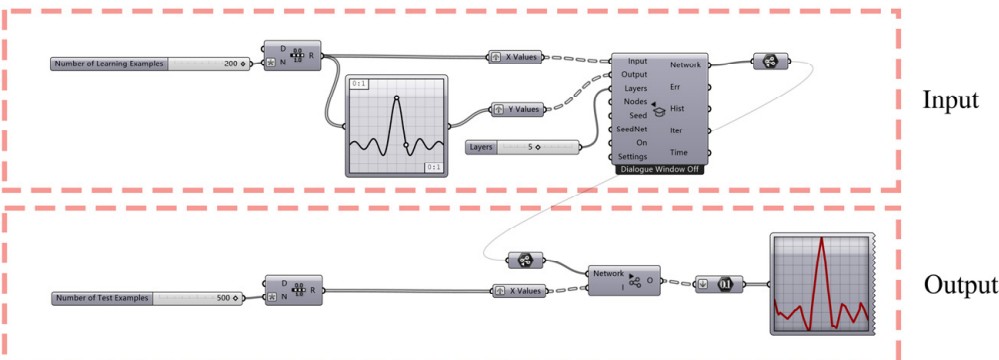

**Figure 5.** ANN battery pack.

SVM is a supervised binary classifier based on the statistical VC dimension theory and the principle of structural risk minimization. Cortes and Vapnik formally proposed a support vector machine (SVM) in 1995 [41]. SVM was first developed from statistical theory and is also known as SVC (support vector classify) for classification problems and SVR (support vector regression) for regression problems. SVMs usually separate samples by a linear function for linearly separable binary classification problems, but there is not only one but an infinite number of straight lines that can separate the data, so the support vector machine corresponds to the straight line that can correctly divide the data with the largest interval. However, such a classifier is still a weak classifier and still cannot solve some linear indistinguishable problems in reality, as shown in Figure 6 in the original space where the data are located. The kernel function in SVM solves this problem by mapping the data from the binary plane to a higher-dimensional space and finding an optimal plane in that space to separate these two types of data. From a mathematical point of view, let the curve in Figure 6, which separates orange and blue, be a circle with the equation $x^2 + y^2 = 1$. The principle of the kernel function is seen as mapping $x^2$ to X and $y^2$ to Y. The equation of the hyperplane becomes X + Y = 1, which makes the nonlinear separable problem in the original space become a linear separable problem in the new space through the mapping effect of the kernel function.

In dealing with the regression problem, the principle is similar to that of SVC. First of all, it is necessary to ensure that the fitted line should reflect all the sample data as much as possible, so the distance between the hyperplane and the farthest sample point in the regression task should be as large as possible, but the situation shown in Figure 7a will occur when the distance is guaranteed to be the largest, so a limit is added to the interval in SVR. The deviation of the model $f(x)$ from y should be less than or equal to $\varepsilon$. The deviation range is also called the $\varepsilon$ pipeline, and the correct fitted curve shown in Figure 7b can be achieved by adding restrictions to the sample data interval. In the case of nonlinear fitting, the same principle as in the classification problem is used to transform the nonlinear fit

into a linear one by mapping the sample data to a higher dimensional space with a kernel function (Figure 8).

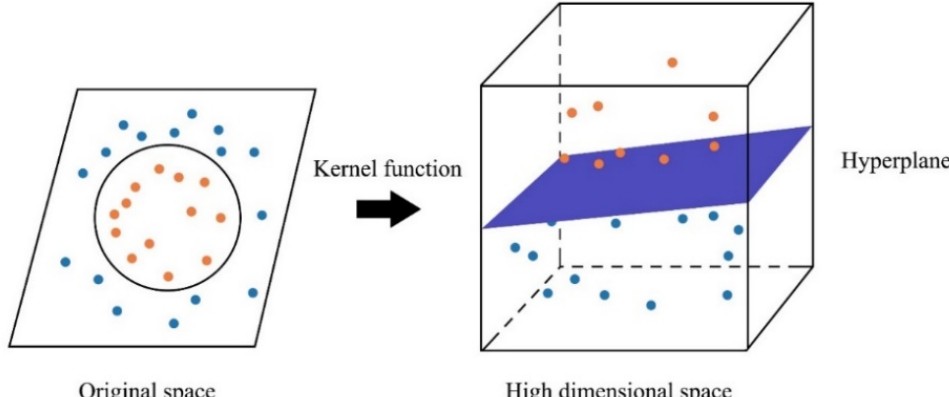

**Figure 6.** The principle analysis of the kernel function in the SVM linear inseparable problem.

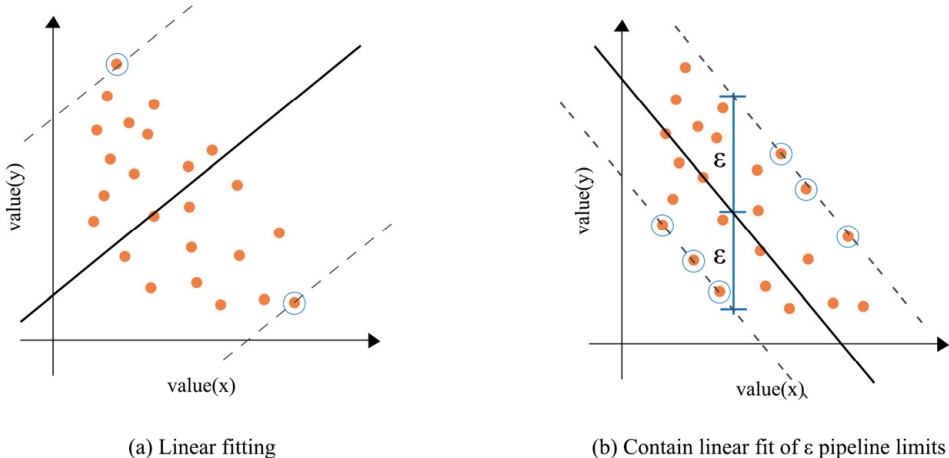

(a) Linear fitting        (b) Contain linear fit of ε pipeline limits

**Figure 7.** SVR linear fitting schematic.

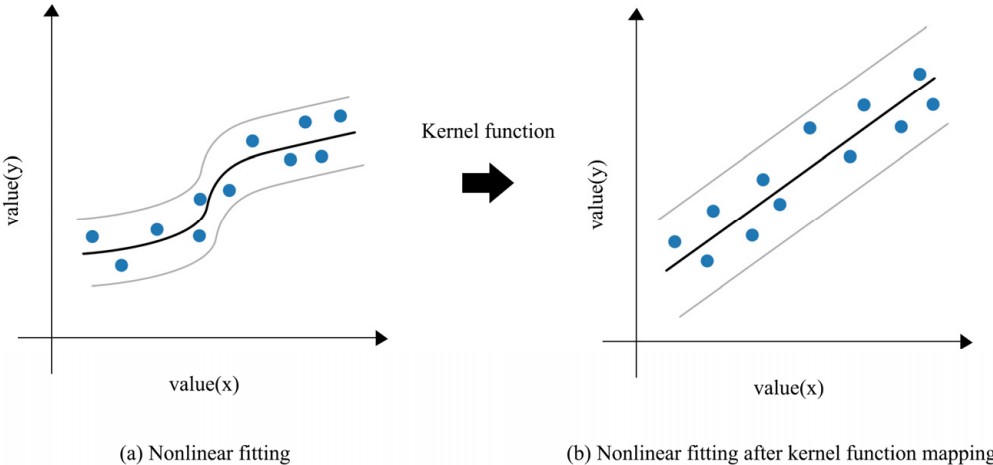

(a) Nonlinear fitting        (b) Nonlinear fitting after kernel function mapping

**Figure 8.** SVR nonlinear fitting schematic diagram.

In Octopus, there is a similar operation logic to ANN, which is also composed of an input module (SVM learning) and an output module (SVM Evaluate). Unlike ANN, the input sample value does not need to be mapped. Because SVM has a strict statistical theory and mathematical foundation, unlike ANN, which needs to rely on the experience

and knowledge of designers, it needs to adjust fewer parameter values, and its operation difficulty is lower than ANN. (Figure 9).

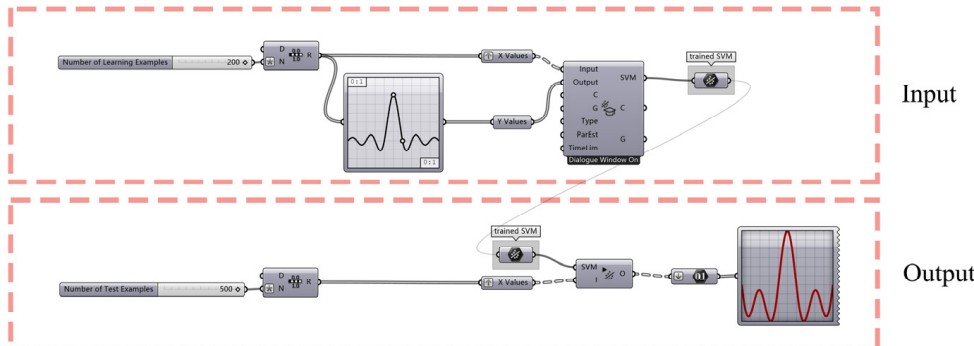

**Figure 9.** SVM battery pack.

3.4.2. Evaluation

The two machine learning algorithms are not superior or inferior in nature, and the more suitable machine learning method should be selected when dealing with different sample data; for example, SVM can be used for small samples with high dimensionality, while ANN can be used for samples with a large number of samples and noisy data. However, in the specific practice process, the operator needs to have strong data sensitivity and certain operating experience, otherwise, it is difficult to judge the nature of the sample data, so this paper proposes a classification attempt, that is, the sample data are imported into two machine learning algorithms for training, and the accuracy of machine learning is evaluated by the following two common model evaluation methods: root mean square error (RMSE) and coefficient of determination ($R^2$) [41]. The machine learning method with the better evaluation grade is used to establish the relationship between the variables requiring higher model accuracy and the performance simulation target value. Among them, the smaller the root mean square error indicates, the better the model prediction, the larger the coefficient of determination, and the higher the model accuracy. Its calculation formula is as follows:

$$\mathrm{RMSE} = \sqrt{\frac{1}{n} \cdot \sum_{i=1}^{n}(y_i - \hat{y}_i)^2} \tag{1}$$

$$\mathrm{R}^2 = 1 - \frac{\sum_{i=1}^{n}(y_i - \hat{y}_i)^2}{\sum_{i=1}^{n}(y_i - \overline{y}_i)^2} \tag{2}$$

where $y_i$ represents the predicted value generated by machine learning, $\overline{y}_i$ represents the average value of $y_i$, and $\hat{y}_i$ represents the simulated value of performance generated by Honeybee calculation. In the "classification attempt" process, 70% of the sample data are usually used for machine learning and 30% for accuracy evaluation. In this paper, the above evaluation method is implemented in GH using a native cell written in the math module (Figure 10).

In addition, if the accuracy of the model is too low for both machine learning methods, it may be caused by factors such as insufficient sample data or improper operation and needs to be returned for verification and modification.

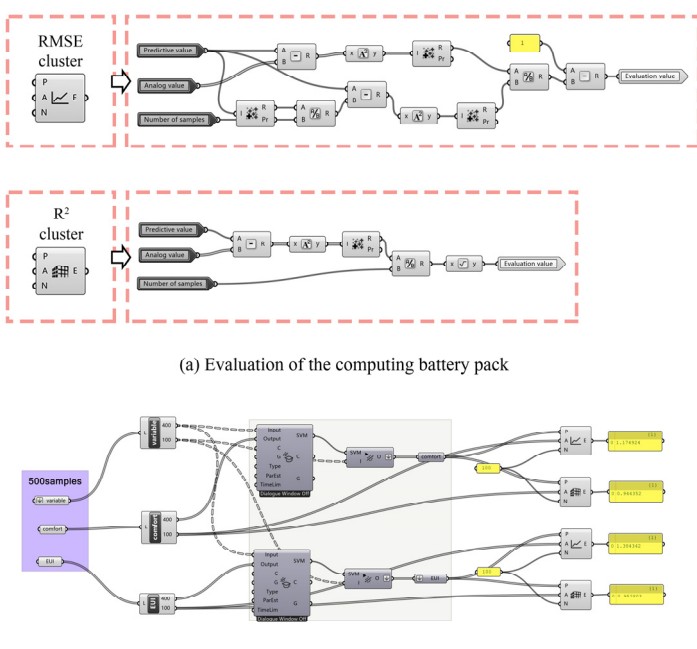

(a) Evaluation of the computing battery pack

(b) Evaluation module usage example

**Figure 10.** Evaluation of modular battery packs and usage examples.

### 3.5. Multi-Objective Optimization

Two important indicators of building energy conservation are energy consumption and comfort. In the optimization screening, multi-objective optimization of the residential building form parameters is performed using the SPEA-2 algorithm equipped with Octopus through a machine learning module in order to find the building form with the lowest energy consumption and the highest comfort level within a specific range of parameters (Figure 11). Since many-objective optimization parameters are contradictory in nature, a solution may be better for one objective but worse for others, so it is necessary to perform a specific analysis in the Pareto solution set [42,43], taking into account the actual situation, e.g., the user of the house spends more time in the house and needs to take more into account the level of indoor comfort, so the Pareto solution set can be chosen to favor the comfort level. For example, in areas where energy is relatively scarce, building energy efficiency is more important, so solutions with lower energy consumption can be selected in the Pareto solution set.

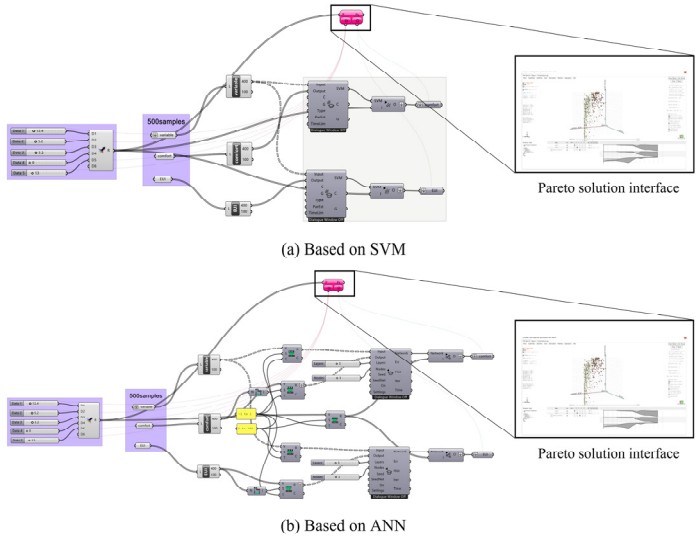

(a) Based on SVM

(b) Based on ANN

**Figure 11.** Multi-objective optimization operation battery pack.

## 4. Experiment and Results

In this section, the above operation process is practically performed so as to verify the effectiveness of this method, and the ANN and SVM algorithms are compared and studied to derive a machine learning approach more suitable for detached residential buildings for practical reference.

### 4.1. Sampling Method Comparison Experiment

With variables $x_1$, $x_2$, which both take values in the range [0, 1], 10 sample data are sampled in the interval using the LHS operation process proposed in this paper with GHs native random number command, respectively, and the generated data are displayed using scatter plots. The comparison of the sampling points in (Figure 12) easily shows that the sampled data after the operation using the above method exhibits the characteristics of uniform LHS sampling distribution, while the sampled data in the traditional random number command has the disadvantages of overlapping data and uneven sampling. This experiment proves the effectiveness of the sampling method in this paper.

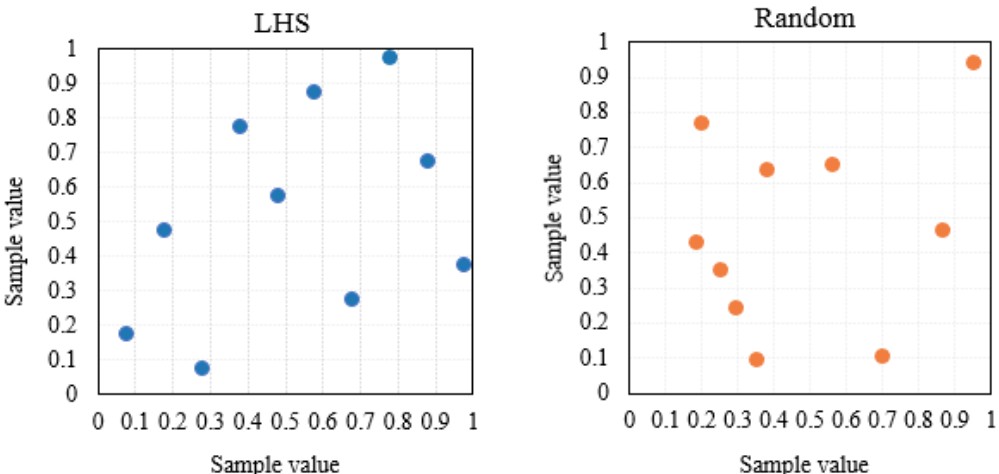

**Figure 12.** Comparison of LHS and Random sampling.

### 4.2. A Comparative Study of Machine Learning Algorithms

Two machine learning algorithms, ANN and SVM, are selected for comparative study. The two algorithms use the same learning samples and test data to ensure the objectivity of the experiment. When generating the sample data, the weather data of Jinan City is selected as an example for the reference independent house building shown in this paper. It adopts the form of a single-layer flat roof. The building is a steel frame structure as a whole, the exterior walls and roof are equipped with insulation layers, and the ventilation rate is set to 0.5 h$^{-1}$. The windows are made of double-layer hollow Low-e glass with an argon-filled air layer in the middle. The thermal transmittance (*U*-value) is 0.8 w/(m$^2$·K), and the solar heat gain coefficient (SHGC) is 0.65. Tables 3 and 4 provide details of the structural and material properties of the building envelope and the internal heat gain.

In this paper, five main parameter variables affecting the form and performance of buildings are set (Table 5), and two important indicators in building energy efficiency design, building energy consumption (EUI) and comfort, are taken as optimization target values. Through the Honeybee plug-in, 500 performance simulations were conducted; 400 samples were used for machine learning, and the remaining 100 samples were used for testing.

**Table 3.** Structural and material properties of the building envelope.

| Construction Component | Layers | Thickness (mm) | Conductivity (W/(m²·K)) | Density (kg/m³) |
|---|---|---|---|---|
| Roof | Roof membrane | 25 | 0.71 | 1856 |
| | Typical insulation | - | - | - |
| | Metal roof surface | 0.8 | 45 | 7824 |
| | Generic insulation | 50 | 0.03 | 43 |
| | HW concrete | 200 | 1.95 | 2240 |
| | Ceiling air gap | 100 | 0.55 | 1.28 |
| | Acoustic tile | 20 | 0.06 | 368 |
| Exterior wall | Stucco | 25 | 0.71 | 1856 |
| | Gypsum board | 15 | 0.16 | 800 |
| | Typical insulation | - | - | - |
| | Gypsum board | 15 | 0.16 | 800 |
| Exterior window | Low-e glass | 6 | 0.99 | 2528 |
| | Argon cavity | 12 | 0.017 | 1.78 |
| | Low-e glass | 6 | 0.99 | 2528 |

**Table 4.** Internal heat gains.

| Source of Internal Heat Gain | Heat Gain | Schedule | |
|---|---|---|---|
| | | **Weekday** | **Weekend** |
| Owner | 100 W/Owner | 00:00–8:00 and 18:00–24:00 | Always at home |
| Lighting | 6 W/m² | 19:00–22:00 | |
| Refrigerator | 150 W | Always on | |
| Television | 60 W | 18:00–22:00 | 8:00–10:00 and 18:00–22:00 |

**Table 5.** Variable parameter settings.

| Variable Name | Value Range | Step |
|---|---|---|
| Width/m | [12, 15] | 0.1 |
| Depth/m | [5, 6] | 0.1 |
| Floor height/m | [3, 4] | 0.1 |
| Building orientation/(°) | [0, 180] | 1 |
| Building window–wall ratio | [0.3, 0.5] | 0.1 |

The first step of training is to determine the number of hidden layers and the number of neurons in the ANN by using the "trial and error" method. The RMSE value reaches the minimum and the $R^2$ value reaches the maximum when the number of hidden layers is 4, and then the RMSE value starts to increase and the $R^2$ value becomes smaller when the number of hidden layers increases. After determining the number of hidden layers to 4 and continuing to test the optimal number of neurons for the ANN, the RMSE and the $R^2$ values start to level off when the number of neurons increases to 7. Continuing to increase the number of neurons does not significantly improve the accuracy of the model but increases the training time. Therefore, in order to balance the training accuracy and training time, a hidden layer of 4 and a number of neurons of 7 were selected to build the neural network, and the RMSE and $R^2$ values of the model were 0.20 and 0.99, respectively. The model has the highest accuracy and takes the least time when the number of hidden layers is 3 and the number of neurons is 10 (Figure 13).

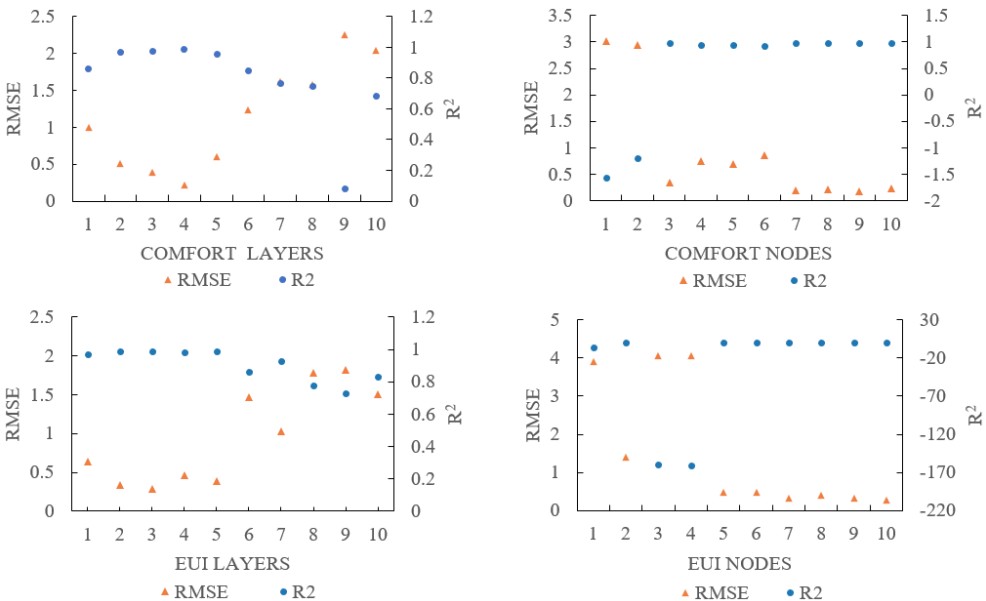

**Figure 13.** Test map of hidden layer and number of neurons.

The same sample data as ANN was used to import SVM for machine learning and use 100 samples for the evaluation test, and the accuracy of the obtained model is shown in Table 6. The comparison of the accuracy of the two machine learning algorithms in Table 6 shows that ANN is better than the SVM algorithm in the prediction accuracy of building energy consumption and comfort for both sample data. The RMSE has the same units as the output variable and can be considered as the average error of the model; the error of SVM in this target value of the annual comfort time is about 3 days, for which the error is within the acceptable range and can be used. SVM algorithm; however, in the prediction of building energy consumption, the error of 1.17 kwh/m$^2$ is too large for this target value because the area of detached houses is generally around 90–200 m$^2$, and it is more appropriate to use the machine learning algorithm of ANN.

**Table 6.** Accuracy comparison of machine learning methods.

| Sample Name | Machine Learning Algorithm | RMSE | R$^2$ |
|---|---|---|---|
| Comfort | ANN | 0.20 | 0.99 |
| | SVM | 1.08 | 0.89 |
| EUI | ANN | 0.29 | 0.99 |
| | SVM | 1.17 | 0.90 |

In previous studies, the accuracy of the model was often used as the only criterion to evaluate the applicability of machine learning algorithms, and only a single algorithm was used in practical applications. This section further compares ANN and SVM algorithms widely used in dealing with regression problems through a simple detached house case on the basis of verifying the feasibility of the proposed method. Considering the complexity and accuracy of machine learning algorithm operation, SVM is simple and fast in operation, so it can be used for year-round comfort prediction with relatively low model accuracy; while ANN is more complicated in operation, but the model is more accurate, so it is more suitable for the model that requires a high forecast of building energy consumption. In summary, two machine learning methods, ANN and SVM, can be integrated and used in the generation of detached residential forms so as to achieve the purpose of efficient operation and accurate prediction.

### 4.3. Building Form Generation Result

The trained machine learning model is used for multi-objective optimization. After 86 iterations, the optimization results converge to form a Pareto front solution set. Figure 14 shows the EUI and comfort values of 500 database cases and Pareto solution sets used for machine learning. Each point in the figure is a solution associated with a set of decision variables representing a design scenario. The Pareto-front solutions yielded better building performance as far as the two optimization objectives are concerned.

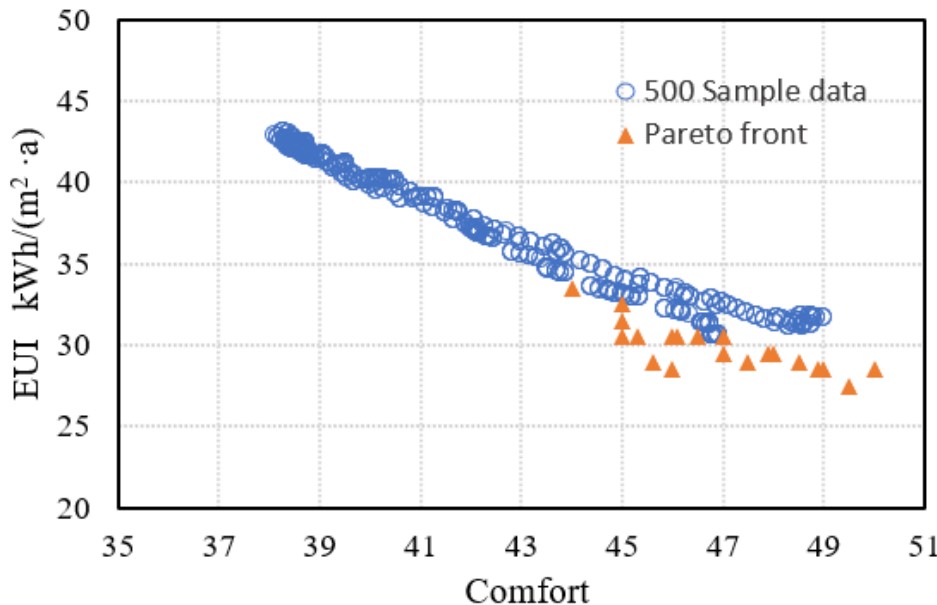

**Figure 14.** EUI and comfort value of 500 sample data and Pareto-front solutions.

Another advantage of the method proposed in this paper is that all the operational processes are completed on one software platform, and the overall consistency is good. The solutions generated from the Pareto solution set can be displayed in Rhino-Grasshopper in real-time. Figure 15 shows nine building form generation cases selected in the Pareto frontier solution set, in which the orange block is the generated building form, the white block is the surrounding environment of the site, and the red line represents the red line of building land. As described in Section 3.5 Multi-Objective Optimization in this paper, the two performance indexes optimized are often contradictory, so they need to be analyzed in detail in practical applications. For example, in the generated case shown in Figure 15, considering the integration of the building and the surrounding sites, Case5, Case6, Case8, and Case9 have improved the comfort of the whole year while ensuring low energy consumption, but they are obviously less integrated with the surrounding sites and exceed the scope of land use. Although Case7 has slightly higher energy consumption than Case4, the comfort of the whole year has been greatly improved; therefore, it is more suitable as a preliminary architectural form solution.

With the support of machine learning and a multi-objective optimization algorithm, this method takes building energy conservation as the goal to explore the optimal building volume. The generated building form model can be intuitively expressed in Rhino-Grasshopper, which can provide data support and intuitive form visual feedback for architects at the conception stage and facilitate further deepening and analysis of the scheme.

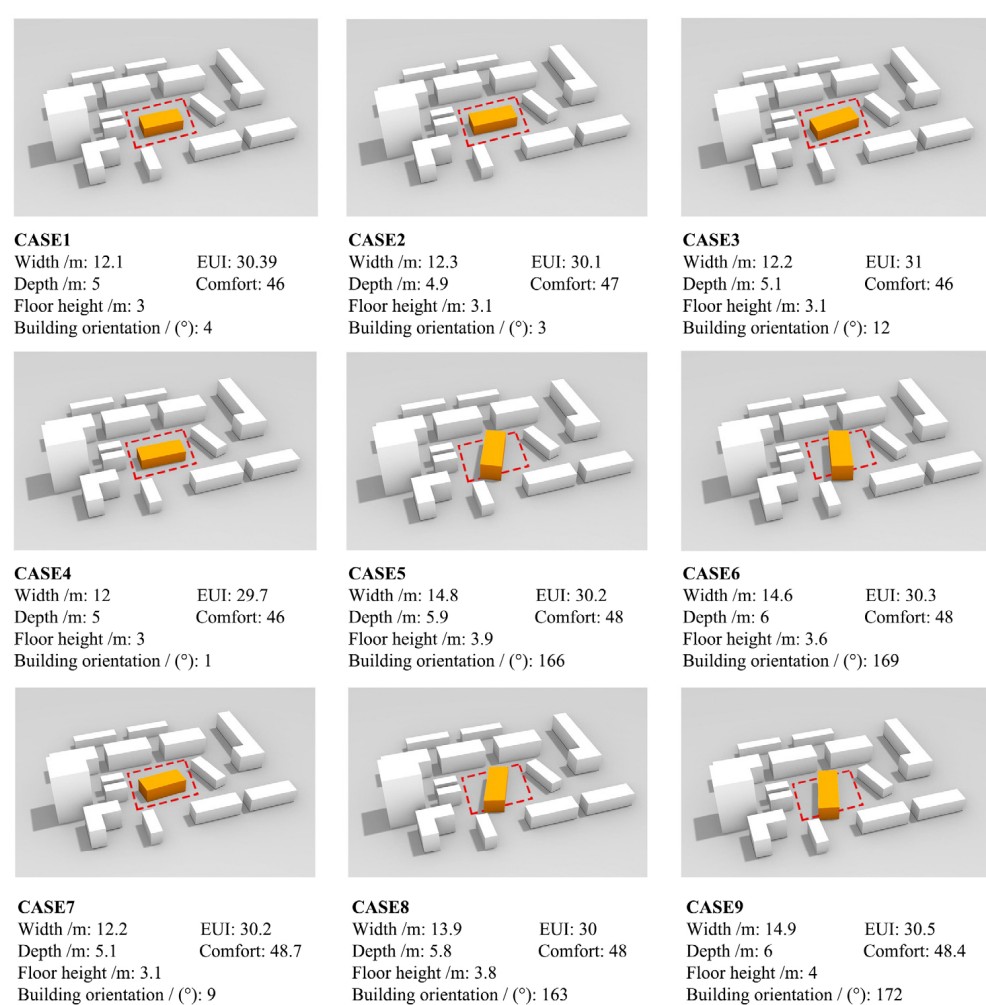

**Figure 15.** Building form generation case (from Pareto solution).

## 5. Conclusions

From the perspective of architects, this paper relies on Grasshopper, a widely used parametric design software in architecture, as a technical platform, and adds innovative modules of data sampling, performance simulation drive, and accuracy evaluation to the Octopus plug-in through visual programming and improves and perfects the algorithm and optimization process it carries. It solves the problem of complex programming and coupling of multiple software programs in the previous study of energy-efficient building design optimization and further improves work efficiency. The applicability of SVM and ANN in different building performance indicators is further analyzed through a residential building case. The experimental results show that ANN and SVM can be used to predict the energy consumption and comfort level of buildings respectively in the design of detached energy-saving houses, which can improve work efficiency while ensuring the accuracy of the model. This method can help architects quickly search for the best building energy-saving form design scheme in the scheme design stage and provide data support and information feedback for architects in design conception and deepening.

Architecture is a unique discipline, combining technology and art. In addition to meeting the requirements of building performance, such as energy-saving optimization, the process of building form generation is also the process of shaping spatial experience and modeling art, so how to effectively balance the relationship between building performance and design aesthetics in the context of future artificial intelligence is a further problem to be solved.

**Author Contributions:** Data curation, H.G.; Formal analysis, D.D. and F.X.; Funding acquisition, D.D.; Investigation, K.D.; Methodology, D.D. and F.X.; Project administration, R.P.; Resources, J.Y.; Software, K.D.; Visualization, J.Y.; Writing – original draft, H.G.; Writing—review & editing, H.G. and R.P. All authors have read and agreed to the published version of the manuscript.

**Funding:** This research was funded by "Teaching Research Project of Wuhan Institute of Technology, grant number x2021019"; "MOE (Ministry of Education in China) Liberal Arts and Social Sciences Foundation, grant number 19YJC760079" and "Graduate Innovative Fund of Wuhan Institute of Technology, grant number CX2020092".

**Data Availability Statement:** Not applicable.

**Conflicts of Interest:** The authors declare no conflict of interest.

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
