# Peer review of "Machine Learning-Based Method for Detached Energy-Saving Residential Form Generation"

_buildings, doi:10.3390/buildings12101504_

Round 1

Reviewer 1 Report

The presented manuscript deals with an interesting topic on an energy efficient building design method based on machine-learing. However, several issues have to be corrected and explained to meet standards required for scientific papers.

General remarks:

1. "Energy-saving" in the title. But Authors don't refer to energy efficient buildings in the manuscript. There were not presented any data of the studied building. Why?

2. In section "Experiment and analysis" authors didn't discuss their findings in relation to other studies. Lack of such a comparison significantly reduces qualilty of their findings. What is the novelty in relation to other authors?

Detailed remarks:

3. line 86: Optimization (not: optimizations)

4. Table 1, Table 2 - Error! Reference ... - please check this error

5. line 164 - this statement is incomplete.

6. line 291: Performance... - not: performance

7. line 313 (and 330): [43] - not: 43, [44] - not: 44

8. line 64 - Error! Reference .. - please check

9. line 420: probably there should be yi and yi instead of yi and y?

10. linie 462: "Five" instead of "5"

11. Table 3: "Width" instead of "width"

12. section 4.2: U-values of partitions, g-glazing, ventialtion flow of a building ?

Reviewer 2 Report

Comments for the paper «Machine Learning Based Method for Detached Energy-saving Residential form Generation”

1: An OK paper with a practical approach on using machine learning models as a tool for better designing of residential buildings.

2: The title of the paper is somewhat confusing, missing the information about any form generation.

3: Missing information about the data used in the paper, the type and amount of data. In machine learning system the most important part is the analysis of the data, and the pipeline for making the machine learning models.

Specific comments to the paper:

1: What do the authors mean with “form generation” with the title of the paper?

2: Introduction:

·         The introduction is ok but try to avoid the use of personal statements like “we”.

·         Include a reference to the Grasshopper platform in line 49?

·         The sentence from line 56 to 61 is too long, and the dot seems to be missing.

·         Include a reference to the GH software in line 74.

3: Literature review:

·         In line 87 “artificial intelligence” (AI) is used instead of machine learning (ML), include the relationship between AI and ML?

·         Table 1 is too wide, redesign, and include proper references.

·         Include a reference to SketchUp and MOEA/D in line 119

·         Start with Machine learning as 2.1 before Optimization?

·         Missing in formation about the data used for making the machine learning models. Include a short description of the data used for the paper, the labels and the number of samples.

·         Missing information about the GAN NN in table 2.

·         Table 2 is too wide and missing most of the references.

·         Include a reference to the “Rhino platform” in line 176

·         ANN consist of several layers and different type of neurons, missing the information about how to control the number of hidden layers and type of neurons in the hidden layers and in the output layer.

·         Missing the information about the input data in Figure 1. What type of data, the type of information in the data, and the amount of information (samples). Data for several years?

·         Line 280 states (Fig, 2) all other references are (Figure x). Be consistent, either Fig. or Figure.

·         Figure 2 is too small and specific for the GH platform? Difficult to understand the figure, explain the buildings blocks.

4: Detached energy-efficient ..

·         Line 313 contains the number 43, should this be a reference?

·         Include any comments that ANN is also good for “non-linear systems” in line 322.

·         Explain the difference of un-supervised and supervised learning in section 2.2, and what is used with ANN and SVM. Confusing with the comment in line 322

·         What do the authors mean with “untrained data” in line 330.

·         The number 44 in line 330 should be a reference?

·         Reference error in line 364.

·         ANN with upper case letters in line 396- and lower-case letters in line 400, be consistent.

·         Include references to RMSE and R^2 in line 412.

·         A lot of focus on RMSE and R^2, missing the same focus on the generation of the ANN (number of hidden layers and type of neurons).

·         User smaller fonts on the RMSE and R^2 formulas (line 417 and 418)

·         The figures are very difficult to understand if you are not familiar with the GH software. Include a short explanation of the building blocks in each figure?

·         Missing information how to analyze the data and divide into training, validation, and test sets.

5: Experiments and Analysis

·         Should the title of the chapter be “Experiments and results”? Missing the analysis part?

·         Any information about the data sets used? Number of labels? Number of samples?

·         The X axis in Figure 12 is the “Sample domin”, what is sample domin?

·         Avoid starting a sentence with the word “And” (line 483)

6: Conclusion

·         Missing any information about the “form generation” from the title of the paper.

Round 2

Reviewer 1 Report

In the revised version ofthe manuscript all remarks have been taken into consideration and the manuscript has undergone significant improvement.

To my knowledge, I can state that I have no further comments regarding scientific content of the manuscript. Only "Error! Reference source not found." should be removed from the tables and references should be formatted following "Instructions for Authors". Currently formatting style of references is incorrect.

Reviewer 2 Report

Thank you very much for your comments and the new version of the paper. This time the paper is much better, comments are:

Table 1 still contains five reference errors.

The header for table 2 seems to be part of the text. Separate the text and the header.

Table 2 still contains five reference errors.

The rest rest of my comments seems to be cover by the extension of the paper.
